# Customer Experience in Open Banking and How It Affects Loyalty Intention: A Study from Saudi Arabia

Ibrahim Mutambik 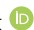

Department of Information Science, College of Humanities and Social Sciences, King Saud University, Riyadh P.O. Box 11451, Saudi Arabia; imutambik@ksu.edu.sa

**Abstract:** The concept of open banking has emerged only recently within the fintech sector, and it is rapidly becoming popular in many regions across the world. Currently, there are very few studies on the relationship between customer experience and intention to use fintech apps, none of which focus on open banking. This relationship is significant for a number of reasons, one of which is the emerging importance of the connection between fintech and an environmentally sustainable economy. This paper seeks to add to our understanding of the factors that shape the customer experience and that determine loyalty levels toward open banking brands and apps. We propose a model in which a number (six) of affective and cognitive factors influence customer experience, which ultimately determines loyalty intention. The model is tested using data collected via a quantitative (survey) methodology involving open banking users in Saudi Arabia. The results show that customer experience is affected by all of the proposed factors (ease of use, perceived value, quality of support, reliability, perceived risk and ability to innovate). These factors, in turn, actively influence the level of customer loyalty. The study contributes to the current literature by identifying the various cognitive and affective determinants of customer experience, which therefore influences loyalty intention in open banking, and provides valuable insights into how both new and established brands should integrate customer experience into promotional and development strategies.

**Keywords:** fintech; open bank; financial technology; customer experience; customer loyalty

## 1. Introduction

Usually defined as the technology used to improve the delivery of financial services, the origins of fintech can be traced back well over a century. In fact, the first significant use of modern-day technology in a financial context is often considered to be the exchange of cotton prices between London and the USA using the world's first transatlantic cable in 1866 [1,2]. However, the term 'fintech' itself only began to gain traction in the late 1990s and early 2000s, as the rise in the use of mobile devices and smartphones enabled the development and adoption of mobile banking apps, while the 2008 global financial crisis fuelled growth of the industry by increasing the demand for non-traditional banking and financial services [3,4]. At present, fintech applications are used across a wide range of industry sectors, ranging from retail banking, investment management and cryptocurrencies to education and fundraising [5].

The advantages of fintech are so significant that the technology is likely to see mass adoption in the next few years. In fact, studies show that the global fintech market is expected to see a compound annual growth rate of approximately 25% between 2022 and 2027, reaching a market value of some USD 324 billion by 2026 [6]. Much of this growth is due to the ability of fintech to facilitate open banking–the process of allowing ordinary citizens to use mobile banking apps to manage their finances simply and securely. As an example of the popularity of open banking, the number of users of the technology in the UK grew from one million to three million between 2020 and 2021 and reached 7 million by February 2023 [7]. Moreover, in the US, almost one in two consumers now

use a fintech solution [8]. Open banking is also an important approach to increasing global financial inclusion, providing unbanked and financially vulnerable individuals with a means of accessing and controlling their own resources [9]. In Saudi Arabia, an Open Banking Framework [10] was launched in 2022 as part of the Saudi Vision 2030, which aims to transform the payments sector in the Kingdom [11] and establish it as a global fintech hub [10]. From a market perspective, the potential of open banking has several important implications. One of these is that the relatively low cost of entry for startups means that competition is high. Not only must new market entrants constantly innovate, but established banks must rethink their business strategies to remain competitive.

Another important implication is that by promoting financial inclusivity, reducing costs and promoting access to green finance, open banking can support fintech's role in transitioning to a low-carbon world [12–14]. This perspective is reflected in the findings of a number of studies. The growing relationship between fintech and sustainability, for example, was demonstrated by Vergara and Agudo [15], while the role of continued use of fintech in delivering sustainability was reported by Ryu and Ko [16]. This paper aims to help us understand the factors that drive continued use. Fintech is also an encouraging environment for startups, which are more sustainable than traditional banks [17].

However, despite the massive potential of fintech and, in particular, open banking, continued growth depends heavily on technological development [18–20], which forms the basis of creating and delivering financial services that meet the dynamic needs of an evolving population [21–23]. This demands that, as competition between brands intensifies, companies must develop their business models to ensure customer satisfaction [3,24,25]. These models will need to use advanced technologies, ranging from DLTs (distributed ledger technologies, such as blockchain) and AI (artificial intelligence) to the IoT (Internet of Things), to provide services that offer faster, more convenient services at lower cost [26,27].

It is not only new and disruptive brands and companies that offer fintech solutions: the traditional financial sector, such as insurance companies and banks, are also adopting fintech approaches to enhance their services [28,29]. However, they face a major challenge in competing with the newer brands, which have built their offering on relevant technology from the ground up and are, therefore, often more agile and responsive to consumer needs [30,31]. In order for financial brands to succeed over the medium and longer term, it is therefore critical that they—whether a traditional or new company—understand the factors that shape the customer experience of a fintech application, which therefore underpin customer satisfaction and resultant loyalty.

Despite the importance of understanding the mechanisms that define and shape the customer experience in fintech, research on customer behaviour in the sector is relatively rare, as the industry is still in the early stages of growth. The result is that the reasons that drive customer adoption of, and loyalty to, fintech apps and services are not fully understood [32,33]. However, as has been noted, such an understanding is key to competitive success in the market, which is rapidly becoming saturated with service providers.

While this is true in most regions of the world, it is particularly true in Saudi Arabia, where the COVID-19 pandemic radically changed customer behaviour in the financial services sector. This resulted in a surge in customer demand for contactless payment services in 2020, leading to the country becoming the largest fintech market in the GCC (Gulf Cooperation Council), with a demographic profile showing almost 70% of the population under 35 years old [34]. In 2022, Saudi Arabia's Fintech Market had a transaction value of USD 48 billion, and it is expected to have a CAGR of more than 12% between 2022 and 2028 [35]. This rapid market expansion means that brands that implement strategies designed to maximise customer loyalty are likely to have a competitive edge. However, there are few studies that explore and analyse the customer experience and its relationship with brand loyalty in fintech, and none which focus on the Saudi context. This research, therefore, addresses a significant gap in the literature. The study explores two principal research questions:

**RQ1.** *What are the main factors which contribute to the consumer experience of open banking fintech apps within Saudi Arabia?*

**RQ2.** *Which of these factors, if any, contribute to brand loyalty in open banking, and to what extent?*

In order to provide insights into these questions, the research develops a model around constructs that form the basis of a number of hypotheses. These hypotheses are tested using a quantitative (questionnaire) methodology. This methodology, as well as results and analysis, are described in detail in the following sections.

## 2. Theoretical Framework

### 2.1. Customer Loyalty

As in all sectors, customer loyalty in banking is defined as the tendency of consumers to continuously purchase products and/or services from a brand, despite the attractions of competitive alternatives [36–39]. Developing and maintaining customer loyalty is a critical issue for businesses, as the cost of acquiring new clients is significantly greater than that of retaining existing ones. Thus, customer relationship management is crucial to business continuation and success. However, building loyalty is a difficult task—this has always been the case, but it is even more true at present, as the increasing use of digital technologies in retail makes purchasing behaviour increasingly volatile [40].

Traditionally, customer loyalty in the banking sector has been strong, and customers have tended to change to a new service provider only after experiencing severe problems [41]. This situation, however, is changing with the emergence of new types of banking institutions and associated services. This, according to Levy and Hino (year), has caused many banks to redirect their marketing strategies from building transactional (short-term) business to developing long-term partnerships, thus, focusing on the development and maintenance of loyalty. Such an approach has various benefits, such as improved retention, reduced service costs, and higher WOM (word-of-mouth) [42], and helps banks to develop and sustain a competitive edge [43,44].

However, the question of how to build loyalty in the banking sector is a complex one, and the issue is further complicated by the fact that open banking is relatively new, and consumer intentions and behaviours are not yet fully understood. This is one of the main aims of this research. The first step in understanding the pathway to consumer loyalty is to understand the concept of the customer journey.

### 2.2. The Customer Journey

The concept of customer experience has evolved significantly in recent decades. Before the digital age, customers had relatively little interaction with sellers: the customer experience tended to be limited to the process of actually purchasing the product. At present, however, the rise of technology, and especially the emergence and ubiquity of smartphones and social media, means that customers often have extended contact with the seller [45,46] before, during and after the purchase process. The customer experience has become more of a journey than a single event, and businesses need to consider every stage of this journey if they are to successfully create a positive experience for consumers. Understanding the critical factors in this journey is increasingly becoming a concern of researchers [47,48].

Acquiring this understanding, however, is not a straightforward process, as the holistic customer experience is highly subjective and interpretive and varies with the individual concerned. In fact, a purchaser's feeling towards a brand is based on the cumulative effect of a number of factors, both cognitive (rational) and affective (emotional). Both of these categories have several sub-dimensions [49–52]. This becomes intuitively clear when we recognise that a typical customer's purchasing behaviour is driven not only by a need to maximise utilitarian aims, such as cutting costs and increasing convenience, but also to maximise hedonic aims, such as gaining pleasure from end-to-end purchasing experience. In other words, the customer experience is essentially a psychological construct [53–56]. However, while trading companies cannot fully control and shape the nature of the cus-

tomer experience, they can influence it to some degree, both positively and negatively, through the use of particular approaches and mechanisms [57]. The precise nature of these mechanisms, and the relationship between them, has been the subject of significant research over the years, though there have been few studies that focus on the open banking sector in fintech.

As we have noted, the customer experience is multidimensional and consists of a range of cognitive and affective factors that are shaped by the consumer's interaction with the brand and its product(s) during the purchasing journey [53,58–60]. Traditionally, most companies think of the user journey as consisting of isolated 'touchpoints', i.e., individual interactions between customers and a business. But to focus on single touchpoints is to miss the holistic nature of the present user journey, which not only extends across a significant period of time but usually involves multiple channels. This means that identifying the user journey in order to understand performance can be a complex task, varying widely from company to company. However, though it may be a complex issue, it is important to address it, as it has been shown that companies that best understand the consumer journey have a greater competitive advantage than those that focus only on touchpoints [61]. In simple terms, while a company may perform fairly well on touchpoints, this does not guarantee good performance on the customer journey [60,62]. Currently, however, it is the user journey that forms the user experience.

This, of course, raises the question of which factors are most important in the user journey and which, therefore, shape the customer experience and form the basis of brand loyalty. The answer to this question varies from sector to sector, and this paper seeks to provide insights for companies in the fintech environment. In this context, the evolving digital realms, together with traditional touchpoints, combine to deliver an individualised customer experience [63–65], and it will be of value for firms to understand the key drivers of positive experiences that underpin loyal behaviours.

*2.3. Customer Experience in Fintech*

Previous research has shown that, within a general retail context, customer experience has a positive relationship with consumer loyalty [66]. It does not necessarily follow from this, however, that the same relationship pertains in the fintech context. In this study, we explore brand loyalty as an outcome of customer experience within the fintech arena.

While there is some research, such as a study of banking in the UK, which suggest that customer experience is positively linked to loyalty [67], it is also true that there are emerging technologies within the fintech sector which mean that findings concerning the traditional banking sector may not apply to the current open banking apps [67–69]. This is mainly because the evolving technological landscape has changed customer expectations and norms in a way that is beyond the reach of traditional banking. A 2023 study by the customer support specialist, Zendesk, for example, showed that 72% of customers want immediate service, while 70% of consumers spend more with companies that offer fluid, personalised and seamless customer experience [70]. Only a few years ago, achieving such objectives would have been beyond the operational scope of most companies.

At present, however, advances in technologies such as AI have made new and higher standards of customer experience possible. Live chat interfaces, for instance, introduced over a decade ago, began to transform customer service into a two-way communication process, with significant effects on trust, satisfaction, WOM (word of mouth) and loyalty [71,72]. More recently, these services have begun to evolve into a service provided by human chat service agents (chatbots), which are designed to communicate with customers using natural language [71]. Other examples, specific to the financial sector, of how digital technology is reforming the boundaries of customer experience are the concepts of the digital wallet, which seeks to create a seamless customer journey by simplifying the processes of e-commerce [73–75], and the robo-advisor, a digital financial 'consultant' that uses mathematical algorithms to provide financial advice or manage investments with minimal human intervention [71,76,77]. Robo-advisors are used to enhance the customer

experience for individuals who have little trading expertise and relatively small sums to invest [78].

More generally, companies across the retail and services sectors are seeking to streamline the customer journey, using advanced technology in many ways. The use of biometrics, for example, is beginning to play a key role in enhancing customer perceptions of service quality in terms of utilitarian values such as simplicity and convenience. Rather than using the relatively clumsy method of passwords or authentication codes to access accounts or services, customers can use a unique biophysical characteristic such as a fingerprint or facial scan [79–81]. However, while the specific technologies and methods used to enhance the customer experience may differ from company to company, they all share the same goal of making their service(s) customer-centric and personalised [79,82].

The model of customer experience in fintech used in this paper is based on two independent but related dimensions: affective and cognitive. The cognitive dimension covers how the consumer thinks and reasons about the brand and its products but is not always a conscious, rational process. The cognitive dimension can have multiple subdimensions that can be affected by unconscious biases [83]. The affective dimension involves the customer's inner feelings and emotions [83], that range from optimism and satisfaction to frustration and anger [53,54]. Affective experiences are related to the consumer's perceptions of fun and pleasure [84].

## 3. The Research Hypotheses and Model

How can we measure customer experience? This, as has been noted, it is essentially a psychological construct that cannot be directly quantified. However, a review of the literature metastudies, e.g., [56,85], suggests that there are six perceived benefits that create the overall customer experience in a general commercial environment: ease of use, perceived value, quality of support, reliability, perceived risk and innovation. This paper evaluates each of these dimensions to assess their contribution to the holistic customer experience in the context of fintech, and specifically open banking.

### 3.1. Ease of Use

This is an important issue in all apps, but it has a particular significance in fintech and open banking. This is for a mixture of utilitarian (cognitive) and emotional (affective) reasons. Due to the nature of a banking app, it is generally used 'on the go', and consumers often have a (utilitarian) consciousness of the time required to complete an activity [86]. The ability to complete shopping (or other) tasks in a timely and convenient manner may lead to a positive customer experience, as individuals are often concerned about the passage of time and tend to make (often unconscious) estimations of time requirements while completing an activity. If these requirements are met, the consumer is usually satisfied, while if they are not met, they can be negatively affected by the experience [87]. However, ease of use also generates a perception of control, which tends to influence the affective dimension of customer experience in a positive way [88].

It is worth noting that ease of use is perceived differently by different consumers. Younger, more 'tech-savvy' users, for example, can be less influenced by the ease of use issue [89], as they (often) adapt more easily to the complexities of usage. For less technically aware consumers, however, ease of use can be critical, and even relatively small difficulties can produce a negative emotion [90]. Despite this, ease of use has at least some influence on the customer experience for all consumers. We therefore hypothesise that:

**H1.** *There is a positive relationship between ease of use and customer experience.*

### 3.2. Perceived Value

The perceived value of a product or service is directly linked to customer satisfaction and can lead to customer loyalty [91,92]. But perceived value is multidimensional. One of the most obvious and most frequently quoted elements of perceived value in the fintech

sector, as well as in other sectors, is financial savings [91,93], but the issue is considerably more complex than this. In fact, as many as 30 separate elements of perceived value have been identified [93]. These elements fall into four categories: life-changing, social impact, emotional and functional. In the fintech context, only the last two (emotional and functional) have a significant impact, including benefits such as functional, i.e., saving/making money, saving time, reducing effort and simplifying life, and emotional, i.e., providing access to services, reducing anxiety and providing pleasure. Perceived value is a key determinant of intention to use online banking, as fintech companies offer a range of end-to-end services that can deliver many of the functional and emotional benefits linked to customer satisfaction. We therefore hypothesise that:

**H2.** *There is a positive relationship between perceived value and customer experience.*

### 3.3. Quality of Support

Customer support consists of a set of processes and mechanisms through which companies provide help to customers when they have a problem [94], and the quality of this support fundamentally affects consumers' perception of the brand. While this is true for all sectors of commerce and retail, it is particularly true in the fintech sector, as customers need to feel that they can depend on the company not only to provide good financial services but to keep their money secure and help them achieve positive outcomes. Because the manner in which support is provided can make the difference between a positive and a negative customer experience, support teams should interact closely with customers, as it is an interaction that can affect the entire consumer journey [95]. We therefore hypothesise that:

**H3.** *There is a positive relationship between the quality of support and customer experience.*

### 3.4. Reliability

Many performance-related aspects of a fintech app can affect the customer experience. Errors or delays in payment processing, for example, can produce a negative perception of a brand, while a reputation for reliability indicates commitment and strengthens customer trust [96]. In fact, it has been demonstrated that trust derived from reliability can be a significant driver of customer loyalty [97–99], while other studies have shown that the perception of reliability in fintech products and services is a primary factor for individuals choosing technology-driven financial services [97,100]. However, there are several factors that can help build the perception of reliability among consumers toward a fintech service provider. The most significant of these factors are security and privacy protection [101], which suggests that consumers are more likely to have a positive fintech customer experience if they perceive that they can rely on the app to provide privacy protection and information security. This finding is in line with that of Suh and Han [102], who reported that authentication and confidentiality play a major role in the perceptions of app reliability in the context of e-commerce. To build a sense of confidence and reliability in consumers, it is also important to include comprehensive and accessible privacy and security policies [103]. Given the above, we hypothesise that:

**H4.** *There is a positive relationship between reliability and customer experience.*

### 3.5. Perceived Risk

In the fintech context, the concept of risk is often considered to be an individual's perception of the chance of suffering loss, either during or after a transaction. In fact, however, risk is a more complex issue than this. A study by Ryu [104] has shown that, instead of considering risk in fintech as a single-dimensional entity, it should be treated as multidimensional, consisting of four components: financial, security, legal and operational. Financial risk is the possibility of financial loss while making a transaction; legal risk is connected to concerns about terms and conditions; security risk is related to loss of privacy

and the abuse of personal information; and operational risk is the possibility of system failure and network problems. Although financial risk is the predominant concern for consumers when using fintech services [54,104], the security and operational dimensions have also been shown to have a significant effect on users' decisions to engage with fintech apps [105]. Cumulatively, these various risk dimensions have been shown not only to act as a barrier to new users [106] but also affect the customer experience and, ultimately, brand loyalty [107,108]. We therefore hypothesise that:

**H5.** *There is a negative relationship between perceived risk and customer experience.*

### 3.6. Ability to Innovate

The ability of a fintech company to enhance existing services, or create and provide new ones using innovative technology, such as artificial intelligence (AI) and machine learning, is a significant factor in medium- and long-term success [109]. This is because innovation can generate positive responses, both affective and cognitive, in consumers, thereby enhancing the customer experience and promoting loyalty [110,111]. When companies gain a reputation for innovation, they are considered to be 'thought leaders' and are in a better position both to attract new customers and to retain existing ones [109–111] while also being able to develop unique and personalised services that are better aligned to consumer needs. This leads to an improved customer journey [110]. It is worth noting that innovation that impacts the customer experience is not confined to the product and service sphere; innovation at the organisational and business-model level can also positively affect consumer perceptions of a company [109]. We therefore hypothesise that:

**H6.** *There is a positive relationship between the ability to innovate and customer experience.*

### 3.7. The Research Model

Customer loyalty is one of the principal goals of any company. This is because when customers purchase on an ongoing basis, they not only buy more over the longer term but spend more per purchase. In fact, studies show that loyal customers spend almost 70% more on products and services than new customers and that it costs five times as much to acquire a new customer than it does to retain an existing customer [112]. For these reasons, among others, 65% of brand executives cite brand loyalty as a key objective of marketing strategy [113]. This raises the key question of how customer loyalty is achieved. One mechanism is through promoting a positive customer experience: research has shown that positive cognitive and affective responses to brand interaction have a positive impact on customer loyalty [20,62,91,114].

This research proposes a model in which customer experience is a fundamental formative element of loyalty intention. The model is based on the concept of the stimulus–organism–response (SOR) model [115]. This proposes that the consumer's environment acts as a stimulus (S), which triggers an (internal) evaluative process in the person (O), which then produces a response (R) [116]. The model suggests that emotions play an important role in the organism's (person's) response to the (environmental) stimulus [117] and that both conscious and unconscious perceptions influence what the person feels [115,117]. In a retail context, the 'stimulus' element consists of the offer of services through a variety of promotional channels [118], while the 'organism' consists of the consumer's (cognitive and affective) processes activated by the stimuli [119]. The 'response' is the outcome of these processes, forms the basis of the customer experience and can be either positive or negative. Positive responses include an intention to purchase or repurchase (loyalty) and an intention to recommend the brand to others [20,62], while negative responses comprise actions such as abandoning a purchase process, failure to repurchase and negative recommendations.

As shown in Figure 1, this study proposes that the 'stimuli' that form the customer experience take the form of various (six) aspects of the user journey: pre-purchase, purchase and post-purchase [120]. These aspects fall within either the cognitive or affective sphere of

response and together impact purchasers' attitudes and behaviours towards loyalty. We therefore hypothesise that:

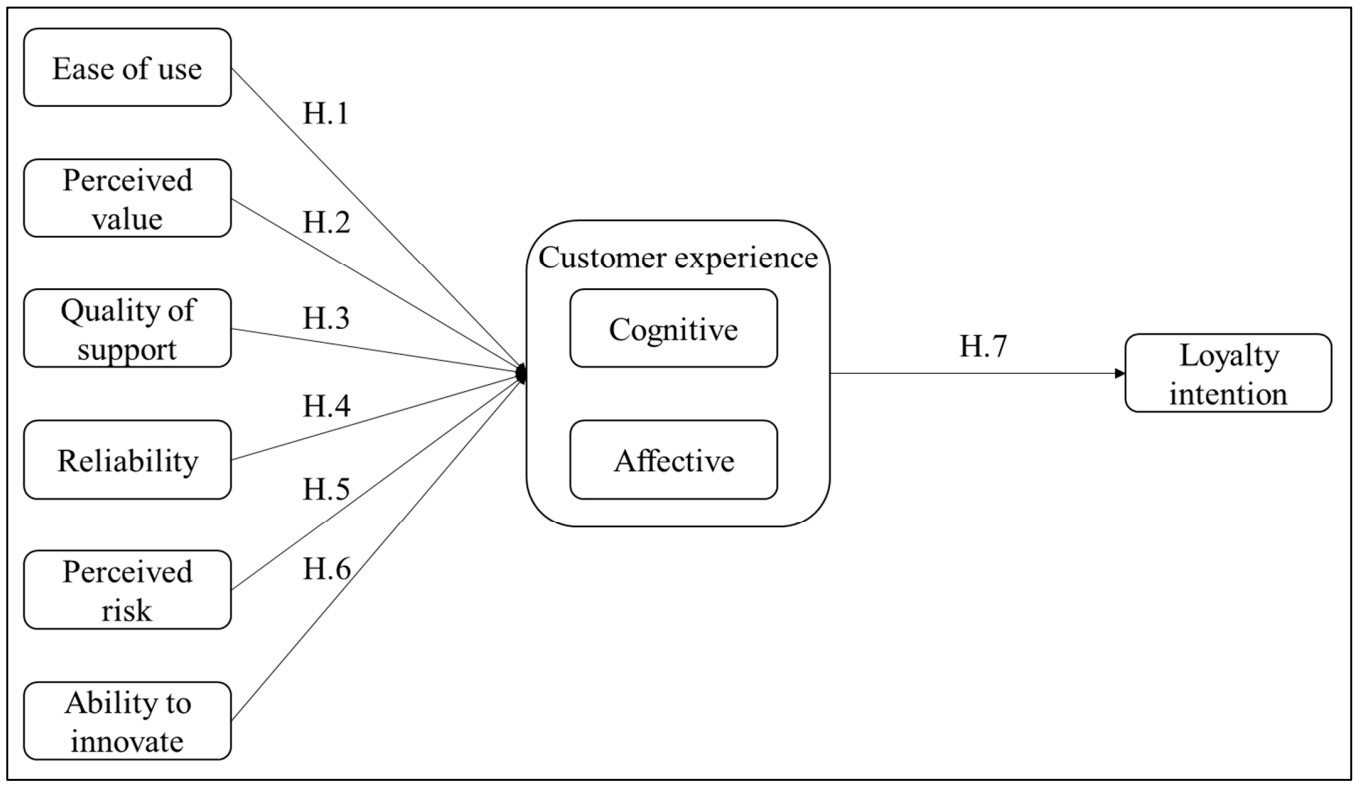

**Figure 1.** The research model.

**H7.** *Customer experience is positively associated with customer loyalty.*

## 4. Research Methodology

### 4.1. Questionnaire Development

There are three main options when designing questionnaires (using questions from other questionnaires, adapting questions used in other questionnaires, or developing customised questions) [121]. In the current case, raw data were produced using a questionnaire adapted from a questionnaire used in several previous studies [61,88,94,122,123]. A 5-point Likert scale was used, and 3 or 4 items were used for each dimension, making a total of 30 items designed to explore the factors that influence the customer experience in a fintech (open banking) context (See Table 1). The questionnaire also captured the demographic profile of the participants in terms of gender, age, employment status and nationality.

Given that the questionnaire was originally adapted from studies in the English language, it was necessary to have it translated into Arabic for those who prefer it. This was performed in order to boost participants' understanding and increase response levels. The translation process required care in order not to change the intended meaning of the original questionnaire [121,124]. A decision was therefore made to use more than one independent translator to translate the text [121]. Comparisons were then made, and the final version was produced by the researcher, ensuring consistency of grammar and syntax in both languages.

In the final step, the decision was made to pilot the questionnaire prior to the main data collection [125]. The aim of the pilot study was to refine the questionnaire in order to decrease the chances that participants would encounter problems in answering the questions and to minimise problems in data recording. Piloting the questionnaire was also important because it (a) helped the researcher to predict the percentage rate of return for

the main study and (b) provided a guide as to how much time was needed to fill out the questionnaire [126]. The pilot exercise consisted of 153 fintech app users, who provided valuable feedback. Some minor changes were made as a result.

**Table 1.** Constructs; items with factor loadings.

| Construct | Item | Factor Loading |
|---|---|---|
| Ease of use | Using banking apps is easy | 0.808 |
| | Banking apps are user-friendly | 0.835 |
| | I have no problems using banking apps | 0.84 |
| Perceived value | Using banking apps save me money | 0.831 |
| | I am happy with the price of using banking apps | 0.829 |
| | Banking apps are good value for money | 0.91 |
| Quality of support | Any issues I have are promptly resolved | 0.808 |
| | It is quick and easy to get support for problems | 0.823 |
| | The support staff are efficient and helpful | 0.833 |
| Reliability | Banking apps are reliable | 0.838 |
| | I have no concerns over security or privacy | 0.918 |
| | The app always works as it should | 0.832 |
| Perceived risk | I do not worry about financial loss while using the app | 0.736 |
| | I consider the risk of using the apps to be low | 0.837 |
| | Using the app is safe | 0.841 |
| Ability to innovate | I feel that banking apps offer innovative services | 0.873 |
| | The apps offer new and creative ways of managing finances | 0.81 |
| | I feel that there is always something new on the way | 0.799 |
| Cognitive experience | Using banking apps is quick and convenient | 0.835 |
| | Using banking apps helps me get what I want | 0.842 |
| | Using banking apps has practical advantages | 0.874 |
| | Using banking apps is cost-effective | 0.828 |
| Affective experience | I find using banking apps a pleasant experience | 0.851 |
| | Using banking apps makes me feel positive | 0.842 |
| | I enjoy using a banking app | 0.868 |
| | I feel better after using a banking app | 0.902 |
| Loyalty Intention | I intend to carry on using the same app(s) | 0.931 |
| | I will recommend my app(s) to others | 0.948 |
| | I don't intend to use other banking apps | 0.912 |
| | My banking app is the best available | 0.927 |

Note: The items were deliberately similar in order to ensure consistency of response [121,127,128].

### 4.2. Sampling and Data Collection

Open banking offers a wide range of benefits, both for individuals and small businesses [11,129]; the demographic profile of users is wide and includes all age and social groups. For this study, a cross-sectional sample, in terms of age, employment status and gender, was used to test our hypotheses. This is important, as younger generations [130] are more likely to have experienced only online banking, whereas older participants are more likely to have experienced traditional banking [130].

In order to obtain data that would provide meaningful insights into our research questions, a number of steps were required. The first of these steps was to identify open banking apps that would provide a representative sample of the Saudi population in terms of both size and demographic profile. After a review of the market, a group of 8 brands was chosen, consisting of a mixture of relatively new and established brands.

The next step was to invite individuals to participate in the study. This was performed by approaching the customer service departments of all eight apps, explaining the purpose of the research, and asking them to distribute the questionnaire to their customers. Ultimately six of the original eight companies agreed to help.

It was agreed with the brands that each of the six companies would send out 750 invitations, making a total of 4500. It was further agreed that the companies concerned would send out invitations to randomly chosen customers; the only criterion for eligibility was that participants must have used the company's open banking services at least once over the previous three months. Over a period of six months, 2633 individuals responded, though only 2590 responses met the study's criteria for eligibility (fully completed questionnaire, use of relevant apps, etc.). Although all invitations were sent to residents of Saudi Arabia, responses were received from consumers of eight nationalities, across both genders and a range of ages and employments. A profile summary of respondents can be found in the Appendix A.

### 4.3. Ethical Issues

All participation in the study was entirely voluntary, and invitations to participate were accompanied by an explanation of the purpose of the research. All potential participants were informed that the study complied with all relevant ethical standards. No direct incentive, financial or otherwise, was offered, though a small contribution to a charity of the participant's choice was promised for every fully completed questionnaire. Further, all participants were given a written assurance that all data collection and analysis were fully anonymised.

## 5. Finding of the Study

### 5.1. Testing the Measurement Model

Confirmatory factor analysis (CFA) is a statistical technique that allows researchers to test the hypothesis that there is a relationship between observed variables and their underlying constructs [131]. This technique was used in the current study to examine model fitness and convergent and discriminant validity. Regarding the goodness of fit indices for the model, the values were all found to be within an acceptable standard range, meeting the criteria recommended by Hair et al. [132] and Hu and Bentler [133]. Table 2 below shows the goodness of fit indices for the structural model.

**Table 2.** Goodness of fit indices.

| Fit Index | Results | Recommended Criteria |
|:---:|:---:|:---:|
| $\chi^2/df$ | 1.652 | $\leq 3$ |
| RMSEA | 0.062 | $\geq 0.06$ |
| IFI | 0.960 | $\geq 0.90$ |
| NNFI | 0.951 | $\geq 0.90$ |

Convergent validity was established by evaluating factor loadings. The results, as shown in Table 1, showed that these loadings ranged from 0.736 to 0.948, which is at an acceptable level as recommended by Hair et al. [132]. Further, as shown in Table 3, the values of average variance extracted (AVE), composite reliability (CR) and Cronbach's alpha (CA) adequately met the required standards [131,132,134]. A discriminant validity test was also made to help ensure that there was sufficient difference between constructs and their metrics [131]. This test compares the square root of the AVE of each construct with the correlation with that construct. The square root of AVE should exceed the correlation values of 0.50 [134]. Again, Table 3 shows that the required standards have been met in the current study.

**Table 3.** Correlations, Cronbach's alpha (CA), composite reliability (CR) and average variance extracted (AVE).

| Factors | CA | CR | AVE | Correlations | | | | | | | | |
|---|---|---|---|---|---|---|---|---|---|---|---|---|
| | | | | 1 | 2 | 3 | 4 | 5 | 6 | 7 | 8 | 9 |
| 1. Ease of use | 0.80 | 0.76 | 0.74 | **0.86** | | | | | | | | |
| 2. Perceived value | 0.77 | 0.75 | 0.72 | 0.73 | **0.85** | | | | | | | |
| 3. Quality of support | 0.79 | 0.74 | 0.69 | 0.30 | 0.38 | **0.83** | | | | | | |
| 4. Reliability | 0.76 | 0.76 | 0.63 | 0.57 | 0.77 | 0.39 | **0.79** | | | | | |
| 5. Perceived risk | 0.79 | 0.73 | 0.66 | 0.69 | 0.72 | 0.32 | 0.71 | **0.81** | | | | |
| 6. Ability to innovate | 0.81 | 0.76 | 0.74 | 0.58 | 0.74 | 0.47 | 0.74 | 0.67 | **0.86** | | | |
| 7. Cognitive experience | 0.77 | 0.75 | 0.72 | 0.71 | 0.49 | 0.41 | 0.471 | 0.7 | 0.39 | **0.85** | | |
| 8. Affective experience | 0.79 | 0.74 | 0.69 | 0.62 | 0.61 | 0.38 | 0.57 | 0.43 | 0.54 | 0.46 | **0.83** | |
| 9. Loyalty Intention | 0.76 | 0.76 | 0.63 | 0.57 | 0.77 | 0.32 | 0.56 | 0.53 | 0.61 | 0.45 | 0.54 | **0.79** |

Note: Square root of AVE shown in bold as the diagonal. CR, AVE and CA were above the cut-off points of 0.7, 0.5 and 0.7, respectively.

Multi-collinearity, which appears if there is a high correlation between the independent variables [112], was also tested. Therefore, the values for the variance inflation factor (VIF) and tolerance were checked to ensure that the former (VIF) was less than three and the latter (tolerance) was over two, as recommended by Hair et al. [112].

*5.2. Common Method Variance and Bias*

As the data were collected from a single source, we tested for common method variance using Harman's single factor test [111]. The common method variance is defined as a systematic error variance that stems from a common method used to measure the constructs of the study [111]. The results indicated that our data are unlikely to be affected by common method variance. We also tested for common method bias, which, while often used interchangeably with common method variance, is conceptually different. Common method bias can occur when independent and dependent variables are measured within one survey using the same (i.e., common) response method [115]. To check for common method bias, we deployed the common latent factor method, as proposed by Field [111]. The results showed that this study was consistent with recommended standards.

*5.3. Findings of the Research Hypotheses*

Structural equation modelling (SEM) was used to examine the psychometric properties of the measurement model, and the hypotheses were tested using the same method. SEM was selected as an analytical tool as it is proven to be effective in the investigation of relationships between constructs [135]. While there are different types of SEM, such as CB and PLS-SEM, the former (CB-SEM) was chosen as it is accepted as most appropriate for theory testing and confirmation, as opposed to exploration and theory building [136].

It can be seen in Figure 2 that ease of use, perceived value, quality of support, reliability, perceived risk and ability to innovate are all related to the customer experience in fintech, explaining 63.7% of the variance. Therefore, H1 to H6 were supported. Further, customer experience in fintech is positively associated with loyalty intentions, explaining 53.4% of the variance. Therefore, H7 was also supported. Table 4 shows the *t*-values and standardised path coefficients of the model.

**Table 4.** Path coefficients and *t*-values for full sample.

| Hypothesis | Path Coefficients | *t*-Statistics | Relationships |
|---|---|---|---|
| H1. Ease of use → Customer experience | 0.37 *** | 8.38 | Supported |
| H2. Perceived value → Customer experience | 0.41 *** | 8.55 | Supported |
| H3. Quality of support → Customer experience | 0.48 *** | 7.81 | Supported |
| H4. Reliability → Customer experience | 0.25 *** | 9.1 | Supported |
| H5. Perceived risk → Customer experience | −0.21 *** | 9.5 | Supported |
| H6. Ability to innovate → Customer experience | 0.22 *** | 8.73 | Supported |
| H7. Customer experience → loyalty | 0.33 *** | 10.05 | Supported |

*** $p < 0.001$.

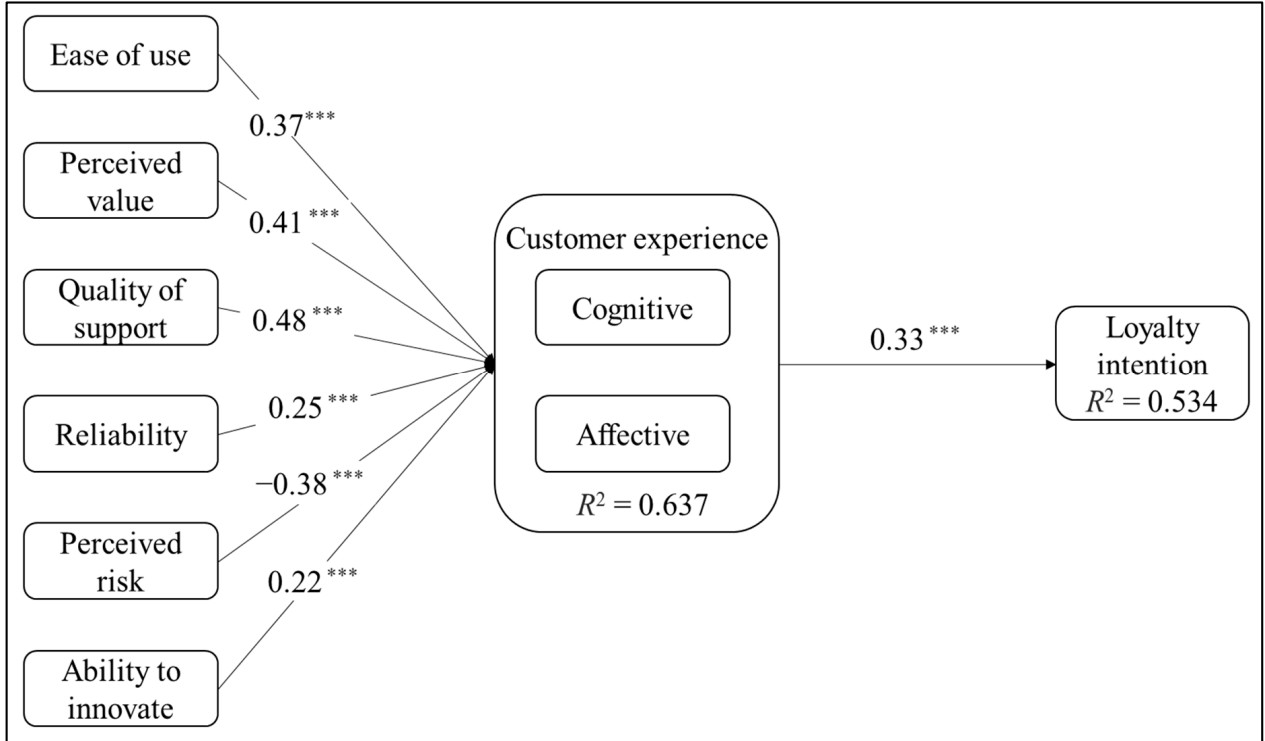

**Figure 2.** The research conceptual model showing empirical results. *** $p < 0.001$.

## 6. Discussion and Theoretical Implications

Although a significant proportion of the population, especially the younger element, are becoming technically aware and familiar with the use of mobile apps, ease of use clearly remains an important factor in shaping the customer experience. This is for a number of reasons. A 2023 study of banking apps in Spain and Portugal, for example, showed a positive connection between ease of use and the level of trust in the app [89], while earlier (2017) research showed that ease of use has an influence on consumers' perception of the actual usefulness of the service provided [137]. Another study of mobile banking services found that ease of use—particularly navigation, clarity of information and availability of key information—is the key differentiator among top-performing mobile apps [53]. While it is tempting to intuitively believe that, as citizens become more technically aware, ease of use becomes less of an issue, this is not the case. The evidence of this study suggests that ease of use remains an important driver of the customer experience for all demographics, and companies should pay particular attention to streamlining the user journey by ensuring functionality such as navigation is clear and simple and implementing advanced techniques such as biometrics to simplify the login process.

Perceived value is also important, as the ability of open banking services to save money and time, through mechanisms such as lower fees while delivering convenience contributes to the customer experience and, ultimately, loyalty [91,93,138]. However, companies vary in the methods used to deliver perceived value: some focus on reducing prices over competitive services, while others seek to increase value by increasing the range and accessibility of services offered [3,139].

While digital technology is becoming increasingly mature and more embedded in the lives of citizens, problems still occur. This can be for several reasons, ranging from system and network malfunctions to user difficulties with the UI (user interface) of the app. When such problems occur, companies that respond promptly and resolve the issue quickly tend to generate a more positive customer experience [140]. Even technically aware users value a good quality of user support, as issues can arise not only from perceived complexities of the app itself but from other network, connectivity and system problems. This is recognised by the industry in general, and service providers increasingly seek to offer support through multiple channels and mechanisms, from human interfaces to digital assistants [141,142]. Customers who have confidence, based on experience, that any issues encountered will be satisfactorily dealt with tend to have higher loyalty intentions. The result is that fast, effective and multi-channel support is a key element of positive customer experience in open banking, as it not only helps customers save time and reduce stress when faced with a problem but generates a feeling among consumers that their money is secure. Companies should, therefore, frequently update customers on new and/or changing support services and ensure that the methods of access to these services are understood.

Reliability in an open banking app is critical to the customer experience for a number of reasons. One of these reasons is connected to the user's primary purpose in using the service, which is to facilitate transactions of one type or another. In order to feel positive about the service provided, the customer must feel that it is reliable; problems, delays or faults in the payment or receipt process will generate a negative feeling of reliability and therefore trust, while fast processing will have the opposite effect [143–145]. Furthermore, studies such as that by Ghani et al. [146] have shown that the perceived reliability of a banking system, including mobile and open banking, significantly influences digital banking effectiveness. However, a feeling of reliability in the consumer also derives from their perception of security [146], which is an issue linked to, but separate from, the dimension of perceived risk. The customer experience is positive when the user relates the service provider to an adequate level of security mechanisms and procedures [147]. Fintech companies should make robust, secure and highly reliable processes and systems a priority to build a sense of trust and confidence in customers.

Although perceived risk has a secondary association with reliability, it has an independent and significant impact on the customer experience. This is principally connected to users' concern about being a victim of online fraud—a criminal activity that represents a major and growing problem for the industry as a whole. According to a report by the FTC (Federal Trade Commission), for example, Saudi Arabian consumers reported losing more than USD 5.8 billion to fraud in 2021, a 70% increase over 2020 [148]. Giving consumers the peace of mind of knowing that they are at minimal risk of suffering financial loss, or having personal data compromised, is, therefore, a key element in the formation of the customer experience. However, perceived risk is a broader issue than just finance, as it also extends to the consumer's perception of operational risk (system or process failure). Consumers who feel at low risk of such events have a more negative customer experience. Companies should, therefore, not only ensure that customers' finances are fully protected as a key priority but should also ensure that they offer the highest levels of data and information privacy, as well as robust systems that have low failure rates.

Innovation is an essential element in a fintech company's value-creation strategy [149]. Most customers expect fintech service providers to deliver unique services which break new technological ground to provide tangible and significant benefits. Customers who see their provider as innovative feel that they benefit from unique services which offer cost and

convenience benefits, as well as access to products that are not available through other (traditional) channels [26,98]. This contributes to the overall customer experience [60,84,140]. In fact, the ability to innovate should be seen as the enabler of the other five dimensions of customer experience. Innovation, at both a technological and service provision level, should be deployed wherever possible to maximise the effect of each of the other dimensions on the consumer experience and, therefore, loyalty. Innovation should not be considered as an aim in itself but rather as a way of meeting customer needs. It is worth noting, however, that research has suggested that the impact of innovation is not the same for all users; it might contribute more to the customer experience of new customers [150]. All dimensions, both affective and cognitive, of customer experience make a significant contribution to loyalty.

Overall, this study has three significant theoretical implications. The first of these concerns is the disassociation of reliability and perceived risk. This implication results from the fact that the study has refined the nature of the component dimensions of customer experience. Although other structurally similar multidimensional models of customer experience exist, which are all complex and holistic [56,60,67,84,140], the model used in this paper proposes dimensions that are more nuanced than most existing models. Thus, for example, we find that reliability and perceived risk, while connected at one level, differ in subtle but important ways.

Another implication of the study derives from insights it provides into the current nature of the formative elements that create the dimensions of customer experience, i.e., it helps us to understand the 'stimuli', which should be used by fintech firms to create specific aspects of the customer experience. As fintech is a rapidly evolving arena, these stimuli change on an ongoing basis, and findings that may have been accurate only a year or two ago may, at present, need to be updated. This has proved to be the case in this research. The study has shown, for example, that perceived value is now about more than just financial savings and that fintech companies can impact the customer experience by innovating at the organisational and business-model level, as well as the technological level.

A third contribution and implication of this research is that while the positive association between customer experience and loyalty has been reported in a general context by other studies [18,58,82,145], this research builds upon these findings by showing that they extend to the specific context of open banking services. This finding has important practical implications, as discussed below.

## 7. Conclusions and Practical Implications for Management

This study set out to provide insights into the factors which determine customer experience of open banking and how these factors contribute to brand loyalty. The findings of the research showed that there are six key factors that shape the customer experience and that this experience has a positive association with brand loyalty. For fintech companies to compete successfully in the rapidly developing market space of open banking, customer loyalty is critical. Loyal customers not only spend more and yield bigger margins [114], but they deliver indirect benefits through WoM (word of mouth) or eWOM recommendations. However, as this study confirms, the antecedent to customer loyalty is customer experience: ensuring a positive customer experience is a prerequisite for the development of loyalty intention. This means that a focus on delivering a positive customer experience should be a top strategic priority for all fintech companies. In order to assess whether these strategic aims are being met, fintech management should continuously assess the customer experience before, during and after the point of purchase [4,68,144].

## 8. Limitations and Future Research

There are a number of factors that limit the generalisability of the findings of this research. One of these limitations is that the study used data collected from a single country (Saudi Arabia). While the sample included a variety of nationalities and, therefore, cultural backgrounds, the focus on a single region might have introduced various biases (e.g., socio-economic and regulatory). Future research using a broader sample is therefore

recommended. Additionally, the study deployed a cross-sectional approach carried out at a specific point in time. However, the needs and preferences of fintech service users are dynamic. Future studies could apply a longitudinal approach to examine the evolution of the customer–brand relationship over a period of time to identify behavioural patterns that could help create a competitive advantage. Additionally, as the main aim of the paper was to explore the relationships between the factors that contribute to consumer experience and loyalty in open banking, the relationships between individual factors that contribute to consumer experience were considered a secondary issue. However, these relationships could contribute to our holistic understanding of loyalty, and further research on this issue would be useful. As open banking is still an emerging concept, and there are very few studies in this area, further research on the nature of specific products and services, as well as a regulatory framework that could potentially help to deliver a positive customer experience, would be valuable. Finally, it was assumed that the group of specific banking apps that agreed to participate in the study was representative of the open banking sector. Further studies could encompass a wider range of fintech service providers.

**Funding:** This research was funded by the Researchers Supporting Project number (RSP2023R233), King Saud University, Riyadh, Saudi Arabia.

**Institutional Review Board Statement:** The study was conducted according to the guidelines of the Declaration of Helsinki, and approved by the Institutional Review Board (Human and Social Research) of King Saud University.

**Informed Consent Statement:** Informed consent was obtained from all subjects involved in the study.

**Data Availability Statement:** Data are available on request due to restrictions of privacy.

**Acknowledgments:** The author would like to extends his sincere appreciation to the Researchers Supporting Project number (RSP2023R233), King Saud University, Riyadh, Saudi Arabia.

**Conflicts of Interest:** The author declares no conflict of interest.

## Appendix A

**Table A1.** Sample profile summary.

|  | Characteristic | Number | Percentage |
|---|---|---|---|
| Gender | Male | 1399 | 54 |
|  | Female | 1191 | 46 |
| Age | 18–24 | 509 | 20 |
|  | 25–49 | 1203 | 46 |
|  | 50+ | 878 | 34 |
| Employment status | Student | 407 | 16 |
|  | Employed | 801 | 31 |
|  | Self-employed | 910 | 35 |
|  | Retired | 472 | 18 |
| Nationality | Saudi | 797 | 31 |
|  | Bangladesh | 343 | 13 |
|  | Egypt | 406 | 16 |
|  | India | 670 | 26 |
|  | Indonesia | 99 | 4 |
|  | Philippines | 108 | 4 |
|  | UK | 83 | 3 |
|  | USA | 75 | 3 |

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
