# Peer review of "Customer Experience in Open Banking and How It Affects Loyalty Intention: A Study from Saudi Arabia"

_sustainability, doi:10.3390/su151410867_

Round 1

Reviewer 1 Report

Thank you to the Author for this manuscript. The publication addresses an important cognitive problem that is relevant both in academic discussion and the business sphere. Like any study of this nature, it has its strengths and limitations. One of the strengths is certainly the intriguing subject matter, which is continuously evolving in scientific research. The research questions are clearly defined, which is a big plus. In terms of weaknesses, it is necessary to specify the research gaps and objectives more clearly. Perhaps stating the main goal and specific objectives would be beneficial. The theoretical framework focuses on the aspect of customer journey. It would be valuable to include aspects related to the title's connection with loyalty. This was lacking in the theoretical section. Not enough attention was given to loyalty issues. The hypotheses are formulated correctly and well justified. Although they may seem obvious, they represent an interesting research area. The model is well constructed. However, it may be worth exploring the relationships between individual components and hypotheses, rather than solely focusing on the relationship with customer experience. The scale is explained, but more attention should be given to the sampling technique. This needs to be addressed. Table 2 should be included as an appendix. The conclusions, limitations, and directions for future research are interesting. The selection of literature also deserves praise as it includes new sources.

Author Response

Dear Reviewer,

Thank you for your comments.

I very much appreciate the time and effort you have given to commenting on my manuscript. The points you have raised are very helpful. I have carefully reviewed the comments and revised the manuscript accordingly. My responses are given in the attached file.

I have also submitted a revised version of my manuscript, and all the changes are marked with track changes.

I look forward to your reply on my revisions. I am happy to make any further changes that will improve the paper. Thank you again for your attention and consideration.

Sincerely yours,

Ibrahim Mutambik,

Reviewer 2 Report

A very interesting study and very on the spot. Nevertheless, these comments need to be considered:

  1.  I was wondering how your paper fits into the aim and scope of this journal, as it seems more like a marketing or a general management paper. I think it is important to link your study to the aims and scope of this journal. I would appreciate it if you could address that by explaining that and probably by including previous studies from this journal in your article.  
  2. It is important you find the gap in this journal and include them in your paper.  
  3. I was just wondering if you could kindly separate the practical and theoretical implications of your paper into separate sections. Preferbally state these before and after the conclusion section.
  4. You have named your article "Customer Experience in open banking How it affects loyalty Intension" I was wondering how you can generalise this study on all customers worldwide whilst you only collect data from a specific country. As researchers, we should be aware that customer behaviour and characteristics may differ from region to region, from one country to another country, and from content to another. To overcome this problem, I suggest you change your article's title and make the study name specific to the country under investigation—for example, Customer Experience in open banking and How it affects loyalty intention; a Study from Saudi Arabia.  
  5. I was just wondering if you could kindly separate your paper's practical and theoretical implications into separate sections. Preverbally state these before and after the conclusion section.

Written in very good English. 

Author Response

(The authors gave the same response as above.)

Reviewer 3 Report

The hypothesis developed are very obvious. Author must use the scales available in literature to map customer experience

The questionnaire developed need to be re worked, all items of the construct are either overlapping or ambiguous for the respondent. There is no clear vision of the author while framing questions. Except for the construct of cognitive experience and Loyalty intentions , others have almost similar items. The author only rephrased the question instead of asking valid question

In methods it  need to be explained why we use SEM. or could we have used PLS SEM

The conclusion also do not bring into any novel insights 

More clarity can be given on financial product innovations and  regulations

English language is Ok.  Readability is fine

Author Response

(The authors gave the same response as above.)

Round 2

Reviewer 3 Report

The revised version can be accepted